# Utilization of Algae Extracts as Natural Antibacterial and Antioxidants for Controlling Foodborne Bacteria in Meat Products

**DOI:** 10.3390/foods12173281

**Published:** 2023-09-01

**Authors:** Gamal M. Hamad, Haneen Samy, Taha Mehany, Sameh A. Korma, Michael Eskander, Rasha G. Tawfik, Gamal E. A. EL-Rokh, Alaa M. Mansour, Samaa M. Saleh, Amany EL Sharkawy, Hesham E. A. Abdelfttah, Eman Khalifa

**Affiliations:** 1Food Technology Department, Arid Lands Cultivation Research Institute (ALCRI), City of Scientific Research and Technological Applications (SRTA-City), New Borg El-Arab 21934, Egypt; tmehany@srtacity.sci.eg; 2Biotechnology and Chemistry Department, Faculty of Science, Alexandria University, Alexandria 22758, Egypt; haneen.samy781@gmail.com; 3Department of Food Science, Faculty of Agriculture, Zagazig University, Zagazig 44519, Egypt; sameh.hosny@zu.edu.eg; 4Department of Food Hygiene and Control, Faculty of Veterinary Medicine, Alexandria University, Alexandria 22758, Egypt; michaelmagdy9090@yahoo.com; 5Department of Microbiology, Faculty of Veterinary Medicine, Alexandria University, Alexandria 22758, Egypt; rasha.gomaa@alexu.edu.eg; 6Department of Food Science and Technology, Faculty of Agriculture, Al-Azhar University, Assiut 71524, Egypt; dr.gamalelsayed@azhar.edu.eg (G.E.A.E.-R.); heshamabdel-mobdy4919@azhar.eud.eg (H.E.A.A.); 7Department of Animal Hygiene and Zoonoses, Faculty of Veterinary Medicine, Alexandria University, Alexandria 22758, Egypt; alaa.m.mansour@alexu.edu.eg; 8Department of Food Science, Faculty of Agriculture (Saba Basha), Alexandria University, Alexandria 21531, Egypt; samaasaleh@alexu.edu.eg; 9National Institute of Oceanography and Fisheries (NIOF), Cairo 11516, Egypt; as.sharkawy@niof.sci.eg; 10Department of Microbiology, Faculty of Veterinary Medicine, Matrouh University, Matrouh 51511, Egypt

**Keywords:** food preservative, algae, natural products, *Padina pavonica*, *Hormophysa cuneiformis*, *Corallina officinalis*, antimicrobial, antioxidant

## Abstract

*Padina pavonica*, *Hormophysa cuneiformis*, and *Corallina officinalis* are three types of algae that are assumed to be used as antibacterial agents. Our study’s goal was to look into algal extracts’ potential to be used as food preservative agents and to evaluate their ability to inhibit pathogenic bacteria in several meat products (pastirma, beef burger, luncheon, minced meat, and kofta) from the local markets in Alexandria, Egypt. By testing their antibacterial activity, results demonstrated that *Padina pavonica* showed the highest antibacterial activity towards *Bacillus cereus*, *Staphylococcus aureus*, *Escherichia coli*, *Streptococcus pyogenes*, *Salmonella* spp., and *Klebsiella pneumoniae*. *Padina pavonica* extract also possesses most phenolic and flavonoid content overall. It has 24 mg gallic acid equivalent/g and 7.04 mg catechol equivalent/g, respectively. Moreover, the algae extracts were tested for their antioxidant activity, and the findings were measured using ascorbic acid as a benchmark. The IC_50_ of ascorbic acid was found to be 25.09 μg/mL, while *Padina pavonica* exhibited an IC_50_ value of 267.49 μg/mL, *Corallina officinalis* 305.01 μg/mL, and *Hormophysa cuneiformis* 325.23 μg/mL. In this study, *Padina pavonica* extract was utilized in three different concentrations (Treatment 1 g/100 g, Treatment 2 g/100 g, and Treatment 3 g/100 g) on beef burger as a model. The results showed that as the concentration of the extract increased, the bacterial inhibition increased over time. *Bacillus cereus* was found to be the most susceptible to the extract, while *Streptococcus pyogenes* was the least. In addition, *Padina pavonica* was confirmed to be a safe compound through cytotoxicity testing. After conducting a sensory evaluation test, it was confirmed that *Padina pavonica* in meat products proved to be a satisfactory product.

## 1. Introduction

Nearly 1 in 10 people worldwide, or 600 million, are expected to get sick from eating contaminated food, and 420,000 people will die as a result. This results in the loss of 33 million DALYs (disability-adjusted life years), or years of good life, foodborne illnesses obstruct socioeconomic progress by taxing health care systems and damaging international trade, tourism, and national economies [1]. Bacterial foodborne illnesses are a frequent occurrence, with several types of foodborne pathogens posing a significant risk. Among them are *Bacillus cereus*, *Staphylococcus aureus*, *E. coli*, *Streptococcus pyogenes*, *Salmonella* spp., and *Klebsiella pneumoniae*. These bacteria can result in a variety of issues, including the production of harmful toxins, triggering mild inflammatory responses, abdominal cramping, and severe diarrhea. In some cases, they can even lead to extensive ulceration of the colonic mucosa [2]. Among the various organisms collaboration with animals, some of these pathogens may gotten to be zoonotic and cause sickness among people, posing a risk to public health and the economy. Animal-derived nourishment items, counting milk, meat, and eggs, are considered fundamental components of human nourishment [3].

Based on this information and projections that Egypt is expected to consume approximately 3.2 million metric tons of meat by 2026 [4], there is a potential problem with meat and meat products being an ideal breeding ground for foodborne diseases and spoiling microorganisms. This is a consequence of their high levels of nutritional content and water activity, which foster growth of these pathogenic microorganisms [5]. Consumers have recently begun thinking of meat and meat products as unhealthy foods. In order to overcome this difficulty, re-formulation is a useful technique for creating customized meat-based products that include substances with specific health-beneficial properties and exclude other traits seen as damaging [6].

As a result, taking precautionary measures to prevent the spread of foodborne illness and ensure the safety of meat products is essential. This can be accomplished through a variety of procedures, including proper storage, handling, and the use of preservation agents. There are two types of preservative substances: natural and synthetic. Most countries prefer chemical substances, which can be synthetic or semi-synthetic, given their affordable price, and antimicrobial effect, and ability to prolong food goods’ shelf lives without compromising their flavor, color, or texture, but it has been demonstrated that these chemical substances have negative health effects [7]. 

Furthermore, each chemical substance provides a specific purpose, with some serving as antibacterial agents, others as antifungal agents, and still others as antioxidants. In order to maintain the safety and quality of food items, the food industry significantly relies on a range of preservatives, for example, potassium nitrate may reduce the oxygen-carrying capacity of blood and potentially cause cancer, calcium benzoate may inhibit digestive enzyme function [8], and sorbic acid, benzoic acid, and their salts can promote the production of substances with mutagenic and carcinogenic effects [9].

Macroalgae, or seaweeds, are an enormous category of varied autotrophs [10]. AlgaeBase now has 172,223 species and infraspecific names, 23,355 images, 67,900 bibliographic articles, and 540,684 distributional information [11]. Marine macroalgae create an abundance of naturally occurring secondary metabolites with varied functions as a result of constant exposure to various biotic and abiotic stimuli that promote the development of such compounds [12]. Algae are particularly noteworthy as photosynthetic organisms that possess exceptional nutritional value, and along with colors like carotenoids, chlorophyll, and phycobilins, bioactive substances include polyphenols, tocopherols, vitamin C, and amino acids that resemble mycosporine [13]. 

Seaweeds include a variety of active substances, such as polyphenols, alkaloids, terpenes, and phlorotannins, in addition to their remarkable concentration of polysaccharides, amino acids, fatty acids, carotenoids, and phycobiliproteins and phycocolloides [14]. These ingredients have strong antioxidant, anti-proliferative, anti-inflammatory, anti-clotting, anti-diabetic, and anti-hepatitis effects [15]. Marine macroalgae represent a potential source of bioactive substances for a variety of uses in the cosmetics, pharmaceutical, agro-food, and, more recently, functional food and chemistry industries [16]. Seaweeds are abundant along Egypt’s Mediterranean, Suez Canal, and Red Sea shores [17], but none are used for commercial purposes, and there is little scientific research on their potential as functional foods and sources of bioactive compounds [18].

Marine algae could be utilized as healthy food sources, nutrition boosters, and preservatives [19]. Additionally, phenolic compounds, aromatic secondary metabolites of marine algae, are important for food’s color and nutritional value [20]. According to reports, seaweeds include significant amounts of dietary fiber (non-starch polysaccharides), essential and non-essential amino acids, minerals, polyunsaturated fatty acids, polyphenolic compounds, vitamins, and other nutrients that are crucial for healthy development [21]. 

A variety of useful metabolites, including vitamins, enzymes, proteins, lipids, carotenoids, polysaccharides, sterols, antibiotics, and a large number of fine compounds, can be found in algae, particularly macroalgae [22]. The secondary metabolite *Padina pavonica*, which was found near Abu Qir Bay in Alexandria, Egypt, is abundant in phenolics, flavonoids, saponins, tannins, and alkaloids. As a result, they are regarded as promising natural candidates for treatment of cancer, diabetes, inflammation, and other diseases. It is advised to continue applied research on such extracts of the examined species for additional therapeutic, medicinal uses, and in vivo studies in the future in order to better understand the bioactive chemicals responsible for the actions [23]. 

The importance of macroalgae as a food source and their abundance of bioactive chemicals are well established. Given that they are palatable, secure, and affordable, these bioactive chemicals can be added to food products for preservation [24]. Determining the qualitative and quantitative phytochemical content of macroalgae has received significant attention. From tropical to temperate seas, *Padina pavonica* is extensively distributed because of its accessibility and ecological characteristics [25]. *P. pavonica* could be used in the production of food [26], and *P. pavonica* (Linnaeus) brown algae with antioxidant, antibacterial, and anticancer properties, is frequently used in soups, salads, and other foods [27]. *Hormophysa cuneiformis* has been shown to have a specific phenolic content, antioxidant properties, and antibacterial properties. As a result, it may be used as a source of biologically active substances [28]. The marine seaweed, *Corallina officinalis*, extracts also maintain strong antimicrobial and antioxidant properties [29,30].

In our study, we aimed to utilize preservative substances that not only ensure safety but also offer multiple beneficial properties, such as antibacterial and antioxidant. We have selected three types of algae species, namely *Padina pavonica* (brown algae), *Hormophysa cuneiformis* (brown algae), and *Corallina officinalis* (red algae), to investigate their effectiveness as food preservatives in various meat products, which are pastirma, beef burger, luncheon, minced meat, and kofta.

## 2. Materials and Methods

### 2.1. Collection of Meat Products and Detection of Pathogenic Bacteria

In Alexandria Governorate, Egypt, a comprehensive study was conducted on 125 meat samples: 25 samples from each product from 5 meat products as follows: pastirma, beef burger, luncheon, minced meat, and kofta, which were obtained randomly from local markets, refrigerated, cut into pieces, and packaged in polyethylene bags then promptly delivered in an ice box for bacteriological examination, and pathogenic bacteria isolated using various selective media were employed to isolate the anticipated harmful microorganisms, according to El-Khawas et al. [31]. 

### 2.2. Bacterial Strains

The study employed six strains of pathogenic bacteria, consisting of 3 Gram-positive strains (*Bacillus cereus* EMCC 1006, *Staphylococcus aureus* EMCC 1351, and *Streptococcus pyogenes* EMCC 1772) and 3 Gram-negative strains (*Salmonella* spp., *Escherichia coli* ATCC 25,922, and *Klebsiella pneumoniae* EMCC 1637). The strains were obtained from the Microbiological Resources Center (MIRCEN), Faculty of Agriculture, Ain Shams University, Cairo, Egypt. Prior to use, the bacterial strains were equipped and customized to a bacterial density of 1 × 10^7^ CFU/mL, following the method outlined by Bahi-Eldin et al. [32].

### 2.3. Algal Materials and Extraction

Total of 3 types of seaweed obtained at Hurghada City, Red Sea Governorate, Egypt: *Padina pavonica* (P.P) (brown algae), *Hormophysa cuneiformis* (H.C) (brown algae), *Corallina officinalis* (C.O) (red algae), and Figure 1. The seaweed specimens were meticulously cleansed of epiphytes, then dehydrated, and transformed into a powdered form. Each type of powdered seaweed was transformed into a lyophilized ethanolic extract (70% ethanol: deionized water *v*/*v*). Recognition of algae was accomplished utilizing the methods described by Salem et al. and Yang et al. [33,34].

### 2.4. Antibacterial Activity

#### 2.4.1. Assessment of the Antibacterial Activity of Algae Extracts by Agar Disk Diffusion Assay

Agar disk diffusion assay was assessed according to Hamad et al. [35] to determine the antibacterial properties of algae extracts against reference strains of pathogenic bacteria purchased from MIRCEN, Faculty of Agriculture, Ain Shams University, Cairo, Egypt). They enriched overnight bacterial cultures on Mueller Hinton Medium (MHM) broth (Oxoid, Cheshire, UK) at 37 °C for 48 h and poured the cultures on MHM plates. Once the plates were dried, they were loaded with each disc that was impregnated with 20 μL of each corresponding algal extract with a concentration of 100 mg/mL each and the plates were incubated at 4 °C for 30 min, followed by incubation at 37 °C/24 h. Inhibitory zones were measured in millimeters to evaluate the anti-pathogenic bacterial activities of the various algae extracts.

#### 2.4.2. Evaluation of Minimum Inhibitory Concentrations MIC of *Padina pavonica* Extract

After assessing antibacterial activity of three types of algae extracts, we focused on the one that demonstrated the highest ability to combat pathogenic bacteria. We then determined the MIC of *P. pavonica* algal extracts, the lowest dose of algal extract that still prevents detectable growth is known as MIC, towards microorganisms that are harmful, following the methodology outlined by Kadaikunnan et al. [36]. To accomplish this, we used varying concentrations of the algal extract that were 100, 50, 25, 12.5, 6.25, and 3.12 mg/mL. Next, we prepared pathogenic bacteria suspensions from cultures that had been grown and adjusted their density to 10^7^ colony forming units (CFU)/mL, as per Bahi-Eldin et al. [32].

### 2.5. Phytochemical Analysis of the Algae Extracts

#### 2.5.1. Total Phenolic Content (TPC) of Algae Extracts 

To evaluate TPC of algae extracts, we used Folin–Ciocalteu test, following the technique outlined by Hamad et al. [37]. We analyzed all samples in triplicate.

#### 2.5.2. Total Flavonoid Content (TFC) of Algae Extracts

We evaluated TFC of algae extracts by a technique outlined by Hamad et al. [38]. 

#### 2.5.3. Assessment of the Antioxidant Activity Diphenyl-1-Picrylhydrazyl (DPPH) Radical Scavenging Capacity

We evaluated algae extracts’ capacity to eliminate DPPH free radicals by modifying the method proposed by Hamad et al. and Catarino et al. [38,39]. First, we prepared a stock solution of each extract in methanol at a concentration of 1 mg/mL. Next, we created serial dilutions of the plant extract by combining 1 mL of each dilution with 1 mL of a methanol solution containing DPPH at a concentration of 1 mg/mL. After incubating the mixture in darkness for 30 min, we measured the absorbance at 517 nm. As a positive control, we used ascorbic acid. The results were expressed as IC_50_, which represents the extract concentration required to reduce 50% of the DPPH. To calculate the IC_50_, we used a non-linear regression algorithm to plot the inhibition percentage against the concentration. The inhibition percentage was calculated using the following formula: DPPH inhibition percentage (%) = [(A of control − A of the sample)/A of control] × 100, where A refers to absorbance.

### 2.6. Safety and Cytotoxicity Assay of Padina pavonica Algal Extract

The potential impact of *P. pavonica* algal extract on PBMCs’ (peripheral blood mononuclear cells) vitality was thoroughly examined. To evaluate cell viability, PBMCs were cultured in Roswell Park Memorial Institute (RPMI) medium. Initially, whole blood was diluted with phosphate-buffered saline (PBS) and formerly carefully layered over an equal volume of Ficoll in a Falcon tube. The mixture was centrifuged for 30 min at 500 rpm to isolate PBMCs. Then, blank wells (150 µL PBS), control wells (150 µL PBMCs), and tested wells (150 µL PBMCs) were placed on a 96-well microtiter plate. Various concentrations of *P. pavonica* algal extracts were then introduced into the tested wells, followed by an incubation period of 24 h in line [40]. Neutral red (150 µL) was added to the wells, and the mixture was incubated at 37 °C for 2 h. The plates were cleaned with a de-staining solution (1% acetic acid: 49% deionized water: 50% ethanol, 150 L/well) after the cells were washed. A T80 UV/VIS spectrophotometer previously set to 540 nm was used to measure absorbance, as mentioned by Ryan and Deci [41]. Using the following equation, *P. pavonica* algal extract inhibition% was computed, and IC_50_ values were obtained through an online calculator [42]. The equation for calculating *P. pavonica* algal extract inhibition% is: Lyophilized HO algal extract inhibition% = 100 − (O.D Control − O.D Treatment O.D)/O.D. Control, where O.D. stands for optical density, control refers to 150 µL PBMCs, and treatment denotes 150 µL *P. pavonica* algal extract. 

### 2.7. Shelf-Life for Padina pavonica Extract as Antibacterial on Beef Burger after Treatment

Prior to the experiment, minced meat was sterilized using ultraviolet light (UV) for 15 min on each side to control the microorganisms present, as suggested by Morsy et al. [43]. The beef burger was prepared according to the method outlined in the Egyptian standard specification for burgers (ESS 1688/1991), as described by Kassem et al. [44]. Fresh beef was transported to the laboratory in an ice box and minced using an electric mincer (Moulinex, 2000 Watt, France) through a 4 mm plate. A mixture of 65 g/100 g minced meat, 20 g/100 g fat, 5 g/100 g soybean, 0.3 g/100 g black pepper, 1.8 g/100 g salt, and 10 g/100 g water was thoroughly mixed for five min in a mixer using a spiral dough hook at medium speed (80 rpm) and passed through a smaller hole plate to ensure homogeneity. The resulting mix was divided into 7 portions, with one portion serving as a negative control without any additives, six portions inoculated with 10^7^ CFU/mL of pathogenic bacteria, including *Bacillus cereus*, *Staphylococcus aureus*, *Streptococcus pyogenes*, *Salmonella* spp., *Escherichia coli*, and *Klebsiella pneumoniae*, serving as positive controls, and 18 portions with the addition of *P. pavonica* extract at 1, 2, and 3 g/100 g, which were also inoculated with 10^7^ CFU/mL of pathogenic bacteria to test their survival. The mixtures were shaped into approximately 50 g cylindrical beef burgers using a commercial forming tool with a 10 cm internal diameter and firmly plasticized film encased to stop moisture evaporation. The burgers were kept in foam plates at 6 °C. Samples were taken at 0, 1, 2, 3, 4, 7, 10, and 15 days of storage for examination of the pathogenic bacteria present. To isolate pathogenic bacteria, we followed a procedure in which 25 g of meat was added to a plastic bag containing 225 mL of 1% peptone water. After homogenizing samples for a minute by a stomacher, we incubated them at 35 °C/24 h. Following this pre-enrichment, we added 1 mL of the mixture to 9 mL of peptone broth and incubated it at 35 °C/24 h. To count pathogenic bacteria, we surface-plated the samples with the proper dilutions and duplicated them on Tryptic Soy Blood Agar (TSBA) medium plates. We conducted three individual replicates of each experiment to ensure accuracy as reported by Ragab et al. and Hamad et al. [45,46].

### 2.8. Assessment of the Acceptability of Beef Burger Fortified with the Padina pavonica Algal Extract

At the Food Technology Department in the City of Scientific Research and Technological Applications in New Borg El Arab, Egypt, a group of 10 experienced evaluators conducted a sensory assessment on a grilled beef burger that was fortified with an extract from *P. pavonica* algae. The objective was to determine the burger’s acceptability as a potential food additive. The evaluators assessed four different groups of samples, including a control group with no treatment and three treatment groups with increasing levels of *P. pavonica* extract [Treatment (1%): beef burger treated with *P. pavonica* extract 1%, Treatment (2%): beef burger treated with *P. pavonica* extract 2% and Treatment (3%): beef burger treated with *P. pavonica* extract 3%. Before evaluation, samples left at room temperature for 10 min Evaluators assessed samples based on odor, taste, color, texture, and overall acceptance, with each criterion being scored on a scale from 1 to 10, where higher ratings denoted a more palatable condition. The sensory attribute data, including standard deviations, were analyzed and recorded according to Hamad et al. [46]. 

### 2.9. Statistical Analysis

SPSS, version 23 (IBM SPSS Statistics for Windows, IBM Corp., Armonk, NY, USA) was used to implement all calculations. For the data analyses, the means standard error (SE) was employed. The Duncan test was employed in one-way analysis of variance (ANOVA), and the probability was regarded as statistically significant when *p* < 0.05.

## 3. Results and Discussion

### 3.1. Detection of Pathogenic Bacteria in Samples

After conducting a bacteriological examination of 125 samples of different meat products (pastirma, beef burger, luncheon, minced meat, and kofta) collected from the local markets in Alexandria Governorate, Egypt, it was discovered that most of the samples were positive and contained pathogenic bacteria. Among the various samples, beef burgers exhibited the highest percentage of positive isolates, with 92%, followed closely by kofta at 88%, minced meat at 84%, and luncheon at 80%. In contrast, pastirma exhibited the lowest percentage of positive isolates at 76%, as detailed in Table 1. This highlighted the hazard of foodborne bacteria in meat products, which affect human health, and the possible source of those bacteria may be from the infected animal or through the slaughtering or contamination during handling. Similar results were obtained by Hassanien [47] who isolated pathogenic bacteria from some meat products, such as kofta, beef burger, and luncheon. Our results confirm those obtained by Bintsis [48] who said that meat-borne diseases are either caused by a number of bacterial infections by contaminating livestock or meat during the meat handling or processing. Our findings are also in line with Ali and Alsayeqh [49], who declared that the influence of meat-borne bacteria on global disease transmission and food safety had a substantial impact on public health.

### 3.2. Antibacterial Activity of Lyophilized Algae Extract

The antibacterial activities of three investigated types of marine algae, namely *Padina pavonica, Hormophysa cuneiformis*, and *Corallina officinalis* by agar disk diffusion against various strains of pathogenic bacteria, *Bacillus cereus*, *Staphylococcus aureus*, *Escherichia coli*, *Streptococcus pyogenes*, *Salmonella* spp., and *Klebsiella pneumoniae*, were evaluated, as illustrated in Table 2 and Figure 2. The results showed that all tested algae extracts have antibacterial effects and inhibit the growth of both tested Gram-positive and Gram-negative strains as followed: *P. pavonica* had the highest antibacterial activity with inhibition zones of 38.83 ± 0.27 and 38.23 ± 0.15 mm against *B. cereus* and *Staph. aureus*, respectively, and the antibacterial activity of *H. cuneiformis* was also notable with inhibition zones of 37.13 ± 0.09 and 34.97 ± 0.15 mm against *K. pneumoniae* and *E. coli*, respectively. Additionally, *C. officinalis* displayed considerable antibacterial activity with inhibition zones of 35.07 ± 0.23 mm against *B. cereus* and 33.17 ± 0.09 mm against *E. coli*, this may be due to marine algae have secondary bioactive substances, such as phenolic and flavonoid compounds, which are known to possess antibacterial properties, this agreed with Al-Saif et al. and Jimenez-Lopez et al. [50,51], who stated that the phenolic contents responsible for the antibacterial activities of algae extract and Jimenez-Lopez et al. and Mohy El-Din and El-Ahwany [51,52] that mentioned the flavonoid composition of algae extracts responsible for the antimicrobial activities. Our results provide promising alternatives of antibacterial agents as antibiotics to overcome the resistant bacteria. This in line with Lee et al. and El Shafay et al. [53,54] who claimed that in order to prevent the emergence of microbial resistance to antibiotics, algae phenolic compounds, or synergistic mixtures of these compounds with other substances, such as fatty acids, halogenated compounds, or terpenes, are a potential new approach. For instance, the antibacterial activity of *P. pavonia* against *K. pneumoniae* in our findings was higher, 34.97 ± 0.09 mm, than that obtained by Al-Enazi et al. [55], who reported 23.40 ± 0.58 mm, and the antibacterial activity of the algae extract against *E. coli* was found to be 18.20 ± 0.63 mm, whereas it was 36.97 ± 0.09 mm against *E. coli* in our study. Another study on *C. officinalis* from the Aegean Sea (Turkey) reported similar results of approximately 32.00 ± 1.73 mm against *E. coli* obtained by Taskin et al. [56], while in our study, it was 33.17 ± 0.09 mm. Therefore, depending on our in vitro antibacterial evaluation results, *P. pavonica* (brown algae) extracts was chosen for further application in meat products against pathogenic bacteria. While Al-Enazi et al. [55] found that extract of *P. pavonica* significantly inhibited tested bacteria by studying (the inhibition zone in mm with correlated concentration of algae extract in mg/mL) as follows: The greatest activities were against *K. pneumonia* (22.60 ± 2.10 mm; 03.90 mg/mL), and against *Staph. aureus* (21.7 ± 0.58 mm; 1.95 mg/mL), while on *Bacillus*, it was (21.7 ± 1.5 mm; 1.95 mg/mL), and against *Strept. Pyogenes*, it was (20.7 ± 1.2 mm; 1.95 mg/mL). Osman et al. [57] declared that *H. cuneiformis* extract at a concentration of 2000 μg/disc (conc. 2) was efficient against both Gram-positive and Gram-negative bacteria, although the concentration of 200 μg/disc (conc. 1) was less effective. Against *E. coli*, it was 6 ± 0.12 mm (conc. 1), 8 ± 0.6 mm (conc. 2), against *Staph. aureus* gave-ve (conc. 1), 9 ± 0.12 mm (conc. 2), and against *Bacillus*, it was 8.5 ± 0.01 mm (conc. 1) and 10.5 ± 0.2 mm (conc. 2). Mofeed et al. [29] also recorded the antibacterial inhibition activity of *C. officinalis* on *E. coli*, *Salmonella*, and *S. aureus*, the results showed that the extract of *C. officinalis* inhibited 100% of *S. aureus* cells at 100 µg/mL concentration, the extract of *C. officinalis* causes the highest inhibition (80%) against *Salmonella*. In conclusion, their results proved that the extract of *C. officinalis* possess an effective antibacterial activity against *Salmonella*, *S. aureus*, and *E. coli* pathogenic bacteria. The variation in inhibition zone size may be due to the concentration of the algae extracts used.

### 3.3. MICs of Lyophilized Padina pavonica Extract

Out of all samples we tested, *P. pavonica* performed best against bacteria. To determine the MIC, we experimented with numerous concentrations of *P. pavonica* extract. The results (Table 3) indicated that the MIC for *B. cereus* and *Staph. aureus* was 3.1 mg/mL, with an inhibition zone of 5.04 ± 0.08 mm and 5.11 ± 0.07 mm, respectively, prooving that they are the most susceptible bacteria to *P. pavonica*. Additionally, the MIC for *Strept. pyogenes* and *E. coli* was 6.2 mg/mL, resulting in an inhibition zone of 6.14 ± 0.09 mm and 4.97 ± 0.09 mm, correspondingly. Finally, we found that the MIC for *Salmonella* spp. and *K. pneumoniae* was 12.5 mg/mL, resulting in an inhibition zone of 6.07 ± 0.07 mm and 7.1 ± 0.09 mm, consistently, that confirmed they are the less sensitive bacteria to *P. pavonica*. The obtained results also highlighted the broad spectrum antibacterial activity of *P. pavonica.* A similar study carried out by Ertürk and Taş [58] showed that the MIC of *P. pavonica* extract was >1.25 mg/mL to *E. coli*, >10 mg/mL to *B. cereus*, >1.25 mg/mL to *Staph. aureus*, >2.5 mg/mL to *Salmonella.* Al-Enazi et al. [55] also found that MIC on *E. coli* 0.00781 mg/mL, *K. pneumoniae* 0.00390 mg/mL, *Bacillus* 0.00195 mg/mL, *Staph. aureus* 0.00195 mg/mL, *Strept. pyogenes* 0.00195 mg/mL.

### 3.4. Total Phenolic and Flavonoids Content of Algae Extracts

Phenolic compounds are frequently present in seaweeds, which contain a range of organic and inorganic compounds that can be beneficial to human health [59]. Algae generally exhibit a greater antioxidant activity because of increased levels of non-enzymatic antioxidant components, including phenols, flavonoids, ascorbic acid, and reduced glutathione [60]. The discovery of new medications and nutritious meals with antioxidant capabilities has spurred interest in marine bio-sources as potential natural sources of bioactive substances. TPC and TFC of *P. pavonica*, *C. officinalis*, and *H. cuneiformis* extracts are demonstrated in Table 4 as: *P. pavonica* (24.13 ± 0.35 and 7.18 ± 0.08 mg GAE/g); (Gallic Acid Equivalent per gram GAE/g); *C. officinalis* (20.03 ± 0.55 and 5.23 ± 0.13 mg GAE/g) and *H. cuneiformis* (15.53 ± 0.29 and 1.83 ± 0.02 mg GAE/g) TPC and TFC; respectively. This was higher than that obtained by Čagalj et al. [61] who said that the TPC of *P. pavonica* ranged from 0.44 ± 0.034 to 4.32 ± 0.15 mg GAE/g and Mannino et al. [62] also determined low TPC in *P. pavonica.* This may be due to environmental challenges and sea temperature as proofed by Mancuso et al. [63] storage, drying, and extraction methods. Čagalj et al. [61] also recorded TFC of *P. pavonica* was 2.25 ± 0.12 mg QE/g (quercetin equivalent per gram QE/g) and said that despite the lack of investigations on the flavonoid content of algae, a few reports have suggested that *P. pavonica* extracts are very flavonoid-rich. Our results for TPC and TFC may be the reason for the antibacterial properties of algae extracts. This is in accordance with with Bansemir et al. [64], who claimed that numerous bioactive substances from marine macro-algae were shown to have antibacterial activity, including alcohols, phenolic hydrocarbons, terpenes, acids, phenols, sulfur-containing compounds, aldehydes, and the skeleton of naphthalene.

### 3.5. Antioxidant Activity and DPPH Radical Scavenging Capacity

During the processing and storage of meat products, the quality of the meat deteriorates due to a process called lipid oxidation. This process results in the formation of primary and secondary oxidation products, reduces the nutritional quality of the meat, and changes its flavor, all of which can pose health risks and result in economic losses due to the production of substandard products [65]. Free radicals play a role in a number of illnesses, such as cancer, AIDS, and neurological diseases. Antioxidants’ scavenging abilities are highly helpful in the management of certain disorders. The most popular approach for evaluating the antioxidant activity of various plant, fungal, or algal extracts is the DPPH assay, which is a sensitive procedure [66]. The DPPH radical has an odd electron, making it a stable free radical that has the maximum absorbance at 517 nm. This electron will be combined by antioxidants with a hydrogen donor to cause a change in color from purple to yellow. The resulting discoloration is stoichiometric in terms of the quantity of electrons consumed [67]. The ability of seaweed extracts to donate electrons and their capacity to scavenge them are what allow the solution containing the DPPH to be bleached [68]. Our results of algae extracts showed antioxidant activity in different concentrations (μg/mL) and compared with standard ascorbic acid in a concentration-dependent manner, the algae extract demonstrated DPPH radical scavenging action. Ascorbic acid’s IC_50_ was discovered to be 25.09 μg/mL, *P. pavonica* was 267.49 μg/mL, *C. officinalis* was 305.01 μg/mL, *H. cuneiformis* was 325.23 μg/mL, as shown in Table 5. This was higher than DPPH scavenging activity obtained by Al-Enazi et al. [55] who found that IC_50_ = 5.59 μg/mL in *P. pavonica*, and the range of *P. pavonica* extracts’ free antioxidant activity against the DPPH radical was 2.33 ± 1.34% to 62.88 ± 3.13% obtained by Čagalj et al. [61], while our results were lower than those obtained by TPC extraction. Antioxidant activity performed by Pinteus et al. [69] found that *P. pavonica* has a DPPH IC_50_ of 338.8 g/mL (338.8 × 10^6^ μg/mL) and a TPC of 44.61 mg GAE/g dry extract. Osman et al. [57] found the IC_50_ of *H. cuneiformis* to be 676.9 µg/mL. This may be due to the methods of inhibition as the measurement conditions, the methods of extraction, or season of collection of algae have a big impact on IC_50_ results. These suggestions also agree with Kandhasamy and Arunachalam [70]. Čagalj et al. [61] claimed that the extraction parameters, solid-to-solvent ratio, and solvent selections in the extraction procedures need to be investigated and optimized in order to identify conditions that would produce most of the targeted compounds while maintaining their biological activity.

### 3.6. Cytotoxicity Effect of Padina Pavonica Extract

Founded on the given data, the cytotoxicity test of *P. pavonica* against PBMCs demonstrates that the extract is not completely safe as concentrations increase. The extract revealed complete inhibition of PBMCs, meaning that it caused cell death at higher concentrations (10,000 µg/mL gave 0% viability followed by 5000 µg/mL gave 4% viability). The lower the concentration, the safer extract will be. The IC_50_ value of 885.8 g/mL indicates that a relatively high concentration of the extract is required to inhibit PBMC cells by 50%, as shown in Table 6. However, it is significant to remember that the IC_50_ value does not always indicate safety, instead representing the concentration at which the extract can cause significant cytotoxicity to cells. As a result, based on the information provided, it would be a bit early to determine that the extract is completely safe to use. More research is needed to assess its safety and potential cytotoxic impacts on other cell types, as well as in vivo studies to determine its safety in humans. Based on the results, *P. pavonica* extract does not show any genotoxicity effect [27]. A similar study carried out by Kosanić et al. [71] confirmed that despite the richness of marine macroalgae in the Adriatic Sea, very little is known about their total phenolic content (TPC) and antioxidant characteristics. Most of these marine macroalgae have not yet been examined for biological activities, *P. pavonica*, from the Adriatic coast of Montenegro, have recently demonstrated that their acetone extracts exhibit antioxidant, antibacterial, and cytotoxic properties. Our results are also consistent with the finding that, after 48 h of treatment, no toxicity was seen in any of the concentrations (0–400 g/mL) of the methanolic and aqueous extract of *Padina* spp. on peripheral blood mononuclear cells (PBMCs) [72].

### 3.7. Shelf Life of Meat Fortified with Padina pavonica Extract

According to the results, the *P. pavonica* extract has antibacterial properties against all the bacteria tested. By increasing concentration, the inhibition increases. Furthermore, as storage time increased, the extract’s antibacterial effect increased. All concentrations of the extract show an effect on all strains, besides, treatments 2 and 3 show complete inhibition in all pathogenic strains by day 10. As shown in Table 7, the extract was most effective against *B. cereus* and least effective against *Strept. pyogenes*. This matched with Ozogul et al. [73] who found that extracts of *P. pavonica* showed potential as antioxidant and antibacterial agents in the sea food sector, extending the shelf life of sea bass fillets by up to 18 days in comparison to the control (12 days). They attributed that to the presence of phenolic chemicals, which may be connected to the studied extracts’ antioxidant and antibacterial activities. Therefore, the plant-based extracts can be used as antioxidant and antibacterial agents in the seafood sector because they were able to increase the microbiological and oxidative stability of treated sea bass fillets when compared to the control. Roohinejad et al. [74] also found that seaweeds are a great source of beneficial active ingredients with antibacterial and antioxidant properties. As previously indicated, phenolics, carotenoids pigments, phlorotannins, and sulfated polysaccharides, to name a few, are the primary phytochemicals responsible for these advantageous qualities. Seaweeds and their extracts may be used in meat products to slow down oxidation reactions and microbiological growth, according to numerous studies. Gullón et al. [6] studied the seaweeds’ and their extracts’ function in maintaining the quality and preventing the rotting of meat products focused on the macroalgae species from which the extracts were obtained. They studies the concentration of the extract or seaweed used, the meat product in which the extract or seaweed was incorporated, and obtained noticeable results. They also included analyses of the incorporation of seaweed or seaweed extracts in meat products and their role in oxidative deterioration.

### 3.8. Sensory Evaluation of Beef Burger Treated through Padina pavonica Extract

According to the data from four groups as control, beef burger fortified with *P. pavonica* extract 1%, 2%, and 3%, the changes in sensory properties are minor, and the general acceptance scores for all extract concentrations remain high. This suggests that at the tested concentrations, adding *P. pavonica* extract to meat may not have a major adverse effect on the sensory properties of the meat, but it also shows a better result when using 3% compared with the control. As a result, our algae extract also acts as a food enhancer, as shown in Table 8. The sensory qualities of the beef burger augmented with *P. pavonica* extract were assessed, and the majority of panelists preferred the *P. pavonica*-fortified version over the control in terms of texture, taste, odor, color, and overall acceptability. 

The overall acceptability of the beef burger strengthened with the *P. pavonica* extract treatment 3% was slightly higher (score of 8.10 ± 0.10) than the control non-treated beef burger (score of 8.05 ± 0.12), while *P. pavonica* extract treatment 1% and 2% (score of 7.95 ± 0.14 and 7.95 ± 0.12, respectively) were slightly lower than both control and treatment 3% groups, even though they were all accepted organoleptically.

When compared to beef burgers supplemented with *P. pavonica* extract treatments 1%, 2%, and 3%, the texture of the control non-treated beef burger (score of 7.90 ± 0.15) received a lower score, compared to scores of 8.05 ± 0.14, 7.95 ± 0.16, and 8.05 ± 0.14, respectively, the superior texture characteristics of the fortified samples may be attributable to the physicosensory capabilities of *P. pavonica* extract, resulting in improved features in the finished product. The color, odor, and appearance of beef burgers fortified with *P. pavonica* extract treatments 1%, 2%, and 3%, were nearly similar to those of the control beef burgers, although the score of fortified groups was still higher compared to the control. A promising, affordable, and secure method for treating beef burgers to regulate antibacterial, antioxidant, and ideal sensory evaluation in the finished product might be to fortify it with *P. pavonica* extract. This agrees with a study by Gullón et al. [6], who found that they are a good natural source of nutrients and biocompounds with a wide range of functions. Edible seaweeds have been suggested to offer intriguing opportunities in the meat industry to make functional foods.

## 4. Conclusions

In conclusion, our study on algae extracts of *Padina pavonica*, *Hormophysa cuneiformis*, and *Corallina officinalis* has demonstrated their antibacterial and antioxidant properties. Therefore, there is a possibility to use algae extracts as food preservatives in meat products. All of the studied seaweed extracts showed a wide range of antibacterial activity, thus, the current study has demonstrated that macroalgae produce antibacterial compounds. Of the extracts tested, *P. pavonica* showed the most significant antibacterial and antioxidant effects against a range of pathogenic strains. Moreover, this extract resulted to be completely safe for human use, with satisfactory features and no negative impact on the sensory properties of meat. Consequently, our findings sustain the prospect of *P. pavonica* and other algae extracts as natural food preservatives in the meat industry that maintain potent antimicrobial activities, making them a future promising source of new antimicrobial as well as preservative agents. This method could provide a viable alternative to traditional preservatives, which might be safe to both the environment and human health. Therefore, additional studies are required to fully comprehend the potential of these algae extracts and their usage in food production. 

## Figures and Tables

**Figure 1 foods-12-03281-f001:**
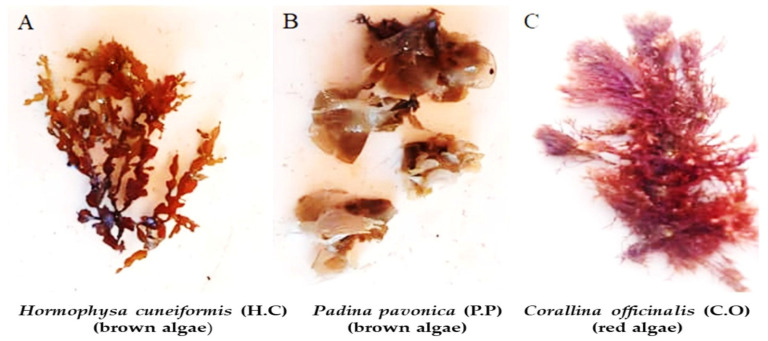
Images of the investigated seaweeds in the present study: (**A**) *Hormophysa cuneiformis* (H.C) (brown algae); (**B**) *Padina pavonica* (P.P) (brown algae), and (**C**) *Corallina officinalis* (C.O) (red algae).

**Figure 2 foods-12-03281-f002:**
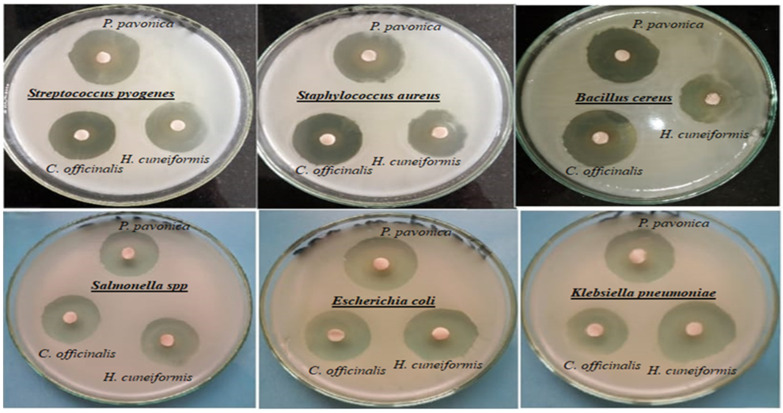
Antibacterial activity of algae extracts *Padina pavonica* (brown algae), *Corallina officinalis* (red algae), and *Hormophysa cuneiformis* (brown algae) against pathogenic bacteria using agar disk diffusion assay.

**Table 1 foods-12-03281-t001:** Incidence of pathogenic bacteria isolates from products collected from different local markets (*n* = 125).

Meat Products	No. of Samples	Positive Samples		Negative Samples	
		No.	%	No.	%
Pastirma	25	19	76	6	24
Beef burger	25	23	92	2	8
Luncheon	25	20	80	5	20
Minced meat	25	21	84	4	16
Kofta	25	22	88	3	12
Total	125	105	84	20	16

**Table 2 foods-12-03281-t002:** Antibacterial activity of lyophilized algae extracts against pathogenic bacteria using agar disk diffusion assay.

Strains	Concentration	Extracts (Inhibition Zone Diameter (mm))		
		*Padina pavonica* (Brown Algae)	*Corallina officinalis* (Red Algae)	*Hormophysa cuneiformis* (Brown Algae)
Gram-positive strains				
*Bacillus cereus* EMCC 1006	100 mg/mL	38.83 ± 0.27 ^c^	35.07 ± 0.23 ^b^	32.5 ± 0.29 ^a^
*Staphylococcus aureus* EMCC 1351		38.23 ± 0.15 ^c^	33.00 ± 0.17 ^b^	30.03 ± 0.20 ^a^
*Streptococcus pyogenes* EMCC 1772		36.00 ± 0.125 ^c^	32.33 ± 0.20 ^a^	33.10 ± 0.06 ^b^
Gram-negative strains				
*Salmonella* spp.	100 mg/mL	33.97 ± 0.09 ^c^	31.23 ± 0.15 ^b^	28.2 ± 0.12 ^a^
*Escherichia coli* ATCC 25922		36.97 ± 0.09 ^c^	33.17 ± 0.09 ^a^	34.97 ± 0.15 ^b^
*Klebsiella pneumoniae* EMCC 1637		34.97 ± 0.09 ^b^	30.20 ± 0.12 ^a^	37.13 ± 0.09 ^c^

Data represented are the means of triplicates ± standard error of means, ^a,b,c^ data in the same column followed by different superscript letters differ significantly (*p* < 0.05).

**Table 3 foods-12-03281-t003:** MICs (mg/mL) of *P. pavonica* extracts against pathogenic bacteria (Inhibition zone diameter in mm).

Strains	Inhibition Zone Diameter (mm) ** with Regard to Each Extract Concentration * (mg/mL) MIC mg/mL						
	100 *	50 *	25 *	12.5 *	6.2 *	3.1 *	MIC
Gram-positive strains							
*Bacillus cereus* EMCC 1006	38.83 ± 0.27 ^f^	24.97 ± 0.03 ^e^	19.10 ± 0.06 ^d^	12.03 ± 0.09 ^c^	8.07 ± 0.04 ^b^	5.04 ± 0.08 ^a^	3.1
*Staphylococcus aureus* EMCC 1351	38.23 ± 0.15 ^f^	23.10 ± 0.06 ^e^	17.17 ± 0.09 ^d^	11.03 ± 0.03 ^c^	7.24 ± 0. 14 ^b^	5.11 ± 0.07 ^a^	3.1
*Streptococcus pyogenes* EMCC 1772	36.00 ± 0.12 ^e^	20.17 ± 0.12 ^d^	14.93 ± 0.12 ^c^	10.23 ± 0.12 ^b^	6.14 ± 0.09 ^a^	ND	6.2
Gram-negative strains							
*Salmonella* spp.	33.97 ± 0.09 ^d^	18.03 ± 0.09 ^c^	11.00 ± 0.06 ^b^	6.07 ± 0.07 ^a^	ND	ND	12.5
*Escherichia coli* ATCC 25922	36.97 ± 0.09 ^e^	21.97 ± 0.20 ^d^	16.10 ± 0.21 ^c^	9.17 ± 0.12 ^b^	4.97 ± 0.09 ^a^	ND	6.2
*Klebsiella pneumoniae* EMCC 1637	34.97 ± 0.09 ^d^	20.14 ± 0.09 ^c^	15.23 ± 0.19 ^b^	7.1 ± 0.09 ^a^	ND	ND	12.5

Data represented are the means of triplicates ± standard error of means, ^a,b,c,d,e,f^ data in the same column followed by different superscript letters differ significantly (*p* < 0.05), MIC; Minimum Inhibition Concentration in mg/mL; * Concentrations of extract in mg/mL; ** Diameter include 5 mm well diameter; ND; Not detected.

**Table 4 foods-12-03281-t004:** Total Phenolic (mg GAE/g) and flavonoid contents (mg catechol/g) of algae extracts.

Extracts	Total Phenolic Content	Total Flavonoids Content
*Padina pavonica*	24.13 ± 0.35 ^c^	7.18 ± 0.08 ^c^
*Corallina officinalis*	20.03 ± 0.55 ^b^	5.23 ± 0.13 ^b^
*Hormophysa cuneiformis*	15.53 ± 0.29 ^a^	1.83 ± 0.02 ^a^

Data represented are the means of triplicates ± standard error of means. ^a,b,c^ Data in the same column followed by different superscript letters differ significantly (*p* < 0.05).

**Table 5 foods-12-03281-t005:** Antioxidant activity and DPPH radical scavenging capacity of the algae extracts.

Concentration μg/mL	Extracts			
	Ascorbic Acid Inhibition	*Padina pavonica* (Brown Algae)	*Corallina officinalis* (Red Algae)	*Hormophysa cuneiformis* (Brown Algae)
25	49.82 ± 0.004 ^d^	3.74 ± 0.02 ^c^	2.06 ± 0.03 ^b^	1.27 ± 0.01 ^a^
50	78.43 ± 0.008 ^d^	8.10 ± 0.06 ^c^	6.78 ± 0.03 ^b^	4.11 ± 0.03 ^a^
75	84.31 ± 0.12 ^d^	12.04 ± 0.02 ^c^	9.19 ± 0.04 ^b^	7.20 ± 0.03 ^a^
100	90.72 ± 0.03 ^d^	15.54 ± 0.04 ^c^	11.85 ±0.03 ^b^	10.11 ± 0.03 ^a^
125	93.29 ± 0.01 ^d^	22.12 ± 0.06 ^c^	14.24 ± 0.05 ^b^	12.60 ± 0.04 ^a^
150	95.29 ± 0.01 ^d^	26.30 ± 0.01 ^c^	20.75 ± 0.05 ^b^	15.15 ± 0.05 ^a^
175	97.39 ± 0.01 ^d^	29.14 ± 0.01 ^c^	25.66 ± 0.02 ^b^	19.08 ± 0.04 ^a^
200	98.37 ± 0.003 ^d^	33.49 ± 0.07 ^c^	28.42 ± 0.06 ^b^	23.88 ± 0.09 ^a^
225	99.26 ± 0.04 ^d^	38.20 ± 0.08 ^c^	34.18 ± 0.05 ^b^	27.61 ± 0.04 ^a^
250	102.20 ± 0.04 ^d^	46.70 ± 0.07 ^c^	39.70 ± 0.01 ^b^	32.33 ± 0.07 ^a^
275	104.10 ± 0.04 ^d^	60.27 ± 0.02 ^c^	45.10 ± 0.01 ^b^	38.29 ± 0.02 ^a^
300	107.30 ± 0.05 ^d^	63.49 ± 0.01 ^c^	60.08 ± 0.01 ^b^	46.16 ± 0.03 ^a^
IC_50_ (μg/mL)	25.09	267.49	305.01	325.23

Data represented are the means of triplicates ± standard error of means, ^a,b,c,d^ Data in the same row between different antioxidant activities followed by different superscript letters significantly differ (*p* < 0.05).

**Table 6 foods-12-03281-t006:** Evaluation of the safety and cytotoxicity assay to *Padina pavonica* extracts on the viability of PBMC cells.

Concentration (µg/mL)	Inhibition %	Viability %
10,000	100	0
5000	96	4
2500	81	19
1250	63	37
625	46	54
312	31	69
156	18	82
78	9	91
39	4	96
19.5	1	99
IC_50_	885.8	

**Table 7 foods-12-03281-t007:** The effect of *Padina pavonica* extract on shelf-life as an antibacterial effect of different concentrations against pathogenic bacteria experimentally inoculated into beef meat stored at 4 °C (mean ± SE).

Strains/Conc. Extract	Inhibition (CFU/g) Storage (Days)							
	0	1	2	3	4	6	8	10
Negative control (1)	0.00	0.00	0.00	0.00	0.00	0.00	0.00	0.00
*Bacillus cereus*								
Positive control	1 × 10^7^	1 × 10^7^	1 × 10^7^	1 × 10^7^	1 × 10^7^	1 × 10^7^	1 × 10^7^	1 × 10^7^
Treatment 1 g/100 g	1.01 × 10^7^ ± 0.01 ^Ca^	1.29 × 10^6^ ± 0.02 ^Da^	1.10 × 10^5^ ± 0.06 ^Cb^	1.76 × 10^4^ ± 0.03 ^Ea^	1.22 × 10^4^ ± 0.01 ^Db^	4.09 × 10^3^ ± 0.05 ^Fc^	0.41 × 10^2^ ± 0.06 ^Bb^	0.00 ± 0.00 ^Aa^
Treatment 2 g/100 g	1.04 × 10^7^ ± 0.03 ^Ba^	1.31 × 10^5^ ± 0.01 ^Ca^	1.05 × 10^4^ ± 0.02 ^Bb^	5.24 × 10^3^ ± 0.03 ^Fc^	2.51 × 10^3^ ± 0.01 ^Ec^	2.10 × 10^2^ ± 0.01 ^Db^	0.00 ± 0.00 ^Aa^	0.00 ± 0.00 ^Aa^
Treatment 3 g/100 g	1.00 × 10^7^ ± 0.003 ^Da^	3.28 × 10^4^ ±0.09 ^Eb^	0.50 × 10^4^ ± 0.004 ^Ca^	3.20 × 10^3^ ± 0.01 ^Eb^	0.40 × 10^2^ ± 0.01 ^Ba^	0.00 ± 0.00 ^Aa^	0.00 ± 0.00 ^Aa^	0.00 ± 0.00 ^Aa^
*Staphylococcus aureus*								
Positive control	1 × 10^7^	1 × 10^7^	1 × 10^7^	1 × 10^7^	1 × 10^7^	1 × 10^7^	1 × 10^7^	1 × 10^7^
Treatment 1 g/100 g	1.00 × 10^7^ ± 0.003 ^Da^	1.00 × 10^6^ ± 0.003 ^Da^	3.10 × 10^5^ ± 0.01 ^Fc^	2.54 × 10^4^ ± 0.03 ^Eb^	0.61 × 10^4^ ± 0.01 ^Ca^	5.31 × 10^3^ ± 0.05 ^Gc^	0.31 × 10^4^ ± 0.01 ^Bb^	0.00 ± 0.00 ^Aa^
Treatment 2 g/100 g	1.01 × 10^7^ ± 0.01 ^Ca^	3.35 × 10^5^ ± 0.04 ^Eb^	1.20 × 10^4^ ± 0.05 ^Db^	0.45 × 10^4^ ± 0.03 ^Ba^	4.48 × 10^3^ ± 0.08 ^Gb^	3.67 × 10^2^ ± 0.07 ^Fb^	0.00 ± 0.00 ^Aa^	0.00 ± 0.00 ^Aa^
Treatment 3 g/100 g	1.04 × 10^7^ ± 0.04 ^Da^	5.31 × 10^4^ ± 0.07 ^Fc^	0.70 × 10^4^ ± 0.01 ^Ca^	5.09 × 10^3^ ± 0.02 ^Ec^	0.60 × 10^2^ ± 0.01 ^Ba^	0.00 ± 0.00 ^Aa^	0.00 ± 0.00 ^Aa^	0.00 ± 0.00 ^Aa^
*Streptococcus pyogenes*								
Positive control	1 × 10^7^	1 × 10^7^	1 × 10^7^	1 × 10^7^	1 × 10^7^	1 × 10^7^	1 × 10^7^	1 × 10^7^
Treatment 1 g/100 g	1.01 × 10^7^ ± 0.05 ^Aa^	7.35 × 10^6^ ±0.03 ^Gb^	6.43 × 10^5^ ± 0.10 ^Fc^	5.60 × 10^4^ ± 0.08 ^Eb^	3.28 × 10^4^ ± 0.04 ^Bb^	8.52 × 10^3^ ± 0.05 ^Hc^	4.14 × 10^3^ ± 0.03 ^Dc^	3.66 × 10^2^ ± 0.05 ^Cb^
Treatment 2 g/100 g	1.01 × 10^7^ ± 0.01 ^Ba^	5.25 × 10^5^ ± 0.03 ^Fa^	4.25 × 10^4^ ± 0.04 ^Ea^	1.22 × 10^4^ ± 0.02 ^Ca^	7.26 × 10^3^ ± 0.03 ^Hc^	6.64 × 10^2^ ± 0.10 ^Gb^	2.16 × 10^2^ ± 0.04 ^Db^	0.00 ± 0.00 ^Aa^
Treatment 3 g/100 g	1.08 × 10^7^ ± 0.06 ^Ba^	8.60 × 10^4^ ± 0.03 ^Gc^	5.40 × 10^4^ ± 0.01 ^Eb^	6.39 × 10^4^ ± 0.03 ^Fc^	2.54 × 10^3^ ± 0.12 ^Ca^	4.11 × 10^2^ ± 0.06 ^Da^	0.00 ± 0.00 ^Aa^	0.00 ± 0.00 ^Aa^
*Salmonella* spp.								
Positive control	1 × 10^7^	1 × 10^7^	1 × 10^7^	1 × 10^7^	1 × 10^7^	1 × 10^7^	1 × 10^7^	1 × 10^7^
Treatment 1 g/100 g	1.07 × 10^7^ ± 0.07 ^Aa^	8.35 × 10^6^ ± 0.04 ^Fb^	7.56 × 10^5^ ± 0.08 ^Eb^	6.98 × 10^4^ ± 0.07 ^Db^	5.32 × 10^4^ ± 0.11 ^Ba^	9.62 × 10^3^ ± 0.06 ^Gc^	6.45 × 10^3^ ± 0.11 ^Cc^	5.37 × 10^2^ ± 0.05 ^Bb^
Treatment 2 g/100 g	1.07 × 10^7^ ± 0.07 ^Ba^	7.36 × 10^5^ ± 0.07 ^Fa^	6.35 × 10^4^ ± 0.04 ^Ea^	3.62 × 10^4^ ± 0.06 ^Ca^	9.07 × 10^3^ ± 0.09 ^Hb^	8.75 × 10^2^ ± 0.10 ^Gb^	4.28 × 10^2^ ± 0.10 ^Db^	0.00 ± 0.00 ^Aa^
Treatment 3 g/100 g	1.01 × 10^7^ ± 0.01 ^Ba^	9.66 × 10^4^ ± 0.09 ^Gc^	8.11 × 10^4^ ± 0.06 ^Ec^	8.50 × 10^3^ ± 0.03 ^Fc^	5.32 × 10^3^ ± 0.07 ^Ca^	7.52 × 10^2^ ± 0.09 ^Da^	0.00 ± 0.00 ^Aa^	0.00 ± 0.00 ^Aa^
*Escherichia coli*								
Positive control	1 × 10^7^	1 × 10^7^	1 × 10^7^	1 × 10^7^	1 × 10^7^	1 × 10^7^	1 × 10^7^	1 × 10^7^
Treatment 1 g/100 g	1.01 × 10^7^ ± 0.01 ^Ba^	6.29 × 10^6^ ± 0.08 ^Fb^	5.36 × 10^5^ ± 0.17 ^Ec^	4.50 × 10^4^ ± 0.03 ^Dc^	2.64 × 10^4^ ± 0.03 ^Cb^	7.31 × 10^3^ ± 0.06 ^Gc^	0.56 × 10^3^ ± 0.02 ^Ab^	2.24 ± 0.09 ^Cb^
Treatment 2 g/100 g	1.00 × 10^7^ ± 0.003 ^Da^	4.25 × 10^5^ ± 0.04 ^Fa^	3.48 × 10^4^ ± 0.03 ^Ea^	0.61 × 10^4^ ± 0.02 ^Ba^	6.30 × 10^3^ ± 0.03 ^Hc^	5.49 × 10^2^ ± 0.04 ^Gb^	0.75 × 10^2^ ± 0.03 ^Cc^	0.00 ± 0.00 ^Aa^
Treatment 3 g/100 g	1.08 × 10^7^ ± 0.06 ^Ca^	7.50 × 10^4^ ± 0.12 ^Fc^	4.11 × 10^4^ ± 0.05 ^Eb^	3.07 × 10^3^ ± 0.15 ^Db^	0.62 × 10^3^ ± 0.08 ^Ba^	1.14 × 10^2^ ± 0.09 ^Ca^	0.00 ± 0.00 ^Aa^	0.00 ± 0.00 ^Aa^
*Klebsiella pneumoniae*								
Positive control	1 × 10^7^	1 × 10^7^	1 × 10^7^	1 × 10^7^	1 × 10^7^	1 × 10^7^	1 × 10^7^	1 × 10^7^
Treatment 1 g/100 g	1.05 × 10^7^ ± 0.03 ^Aa^	7.67 × 10^6^ ± 0.09 ^Fb^	7.22 × 10^5^ ± 0.07 ^Eb^	6.30 × 10^4^ ± 0.15 ^Db^	4.53 × 10^4^ ± 0.12 ^Ba^	9.31 × 10^3^ ± 0.06 ^Gc^	5.90 × 10^3^ ± 0.05 ^Cc^	4.61 ± 0.15 ^Bb^
Treatment 2 g/100 g	1.00 × 10^7^ ± 0.01 ^Ba^	6.45 × 10^5^ ± 0.03 ^Fa^	5.21 × 10^4^ ± 0.05 ^Ea^	2.49 × 10^4^ ± 0.19 ^Ca^	8.24 × 10^3^ ± 0.09 ^Gb^	8.31 × 10^2^ ± 0.07 ^Gb^	3.53 × 10^2^ ± 0.06 ^Db^	0.00 ± 0.00 ^Aa^
Treatment 3 g/100 g	1.00 × 10^7^ ± 0.00 ^Ba^	9.30 × 10^4^ ± 0.06 ^Gc^	7.30 × 10^4^ ± 0.06 ^Eb^	8.14 × 10^3^ ± 0.03 ^Fc^	4.57 × 10^3^ ± 0.09 ^Ca^	6.50 × 10^2^ ± 0.06 ^Da^	0.00 ± 0.00 ^Aa^	0.00 ± 0.00 ^Aa^

Data represented are the means of triplicates ± standard error of means. Pathogenic bacteria counts are in (Log10 CFU/g). ^A,B,C,D,E,F,G,H^ Data in the same column between the same treatment at different storage periods followed by different superscript letters significantly differ (*p* < 0.05). ^a,b,c^ Data in the same column between different treatments at the same storage periods followed by different superscript letters significantly differ (*p* < 0.05).

**Table 8 foods-12-03281-t008:** Acceptability of un−inoculated beef burger fortified with *Padina pavonica* extract depending on sensory attributes.

Treatment/Group	Sensorial Properties Mean (SD)					
	Color	Odor	Taste	Texture	Appearance	Overall Acceptability
Control	7.70 ± 0.15 ^b^	7.90 ± 0.12 ^a^	8.00 ± 0.11 ^a^	7.90 ± 0.15 ^ab^	7.85 ± 0.13 ^a^	8.05 ± 0.12 ^a^
Treatment 1%	7.70 ± 0.13 ^b^	7.80 ± 0.15 ^a^	7.90 ± 0.12 ^a^	8.05 ± 0.14 ^a^	7.65 ± 0.13 ^b^	7.95 ± 0.14 ^a^
Treatment 2%	7.80 ± 0.15 ^a^	7.90 ± 0.15 ^a^	7.95 ± 0.17 ^a^	7.95 ± 0.16 ^a^	7.85 ± 0.18 ^a^	7.95 ± 0.12 ^a^
Treatment 3%	7.90 ± 0.19 ^a^	7.90 ± 0.12 ^a^	8.05 ± 0.14 ^a^	8.05 ± 0.14 ^a^	7.80 ± 0.15 ^a^	8.10 ± 0.10 ^a^

Control: Beef burger without any treatment, Treatment (1%): Beef burger treated with *P. pavonica* extract 1%, Treatment (2%): Beef burger treated with *P. pavonica* extract 2%, Treatment (3%): Beef burger treated with *P. pavonica* extract 3%, ^a,b^ Data in the same column between different treatments followed by different superscript letters significantly differ (*p* < 0.05).

## Data Availability

The data presented in this study are available and contained within the article.

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
