# Peer review of "Utilization of Algae Extracts as Natural Antibacterial and Antioxidants for Controlling Foodborne Bacteria in Meat Products"

_foods, 2023, doi:10.3390/foods12173281_

Round 1

Reviewer 1 Report

The manuscript Foods-2547414 entitled " Utilization of Algae Extracts as Natural Antibacterial and Antioxidants for Controlling Foodborne Bacteria in meat products”

Overall, the paper is well written and very well explained.

I have few suggestions for improvement.

Why the authors did not study lipid oxidation by measuring TBARS values. It was very important to determine the oxidative stability of the meat products treated with algae extracts. Additionally, TVBN is also an important indicator in such studies missed by the authors. I will recommend to add the relevant data, if its available. It will improve quality.   

Abstract: Please modify abstract according to the Journal’s guidelines. It should be in a single paragraph without sub headings. The main content of the abstract should include the briefly purpose of the research, the principal result and major conclusion. The abstract, in the present form, is not adequate. Please add specific results here not generalized. Additionally, (i) the authors must state the revised justification in the abstract to support the study; (ii) Conclusion in the abstract is too general, authors shall state the important finding in this study before going into the general application

Introduction: The introduction section does not explain what was done on topic analyzed. Please include the relevant information about the algae extract antimicrobial effects specifically related to its use in meat products.

Figure 1: Please improve the figure pixels and quality. Also remove background to improve their quality.

2.5.1. Total Phenolic content (TPC)

No need to give details of already well-established methods. Relevant reference is enough. Delete the following paragraph, it is unnecessary

“To begin, we added 0.1 mL of reconstituted extract to 0.1 mL of Folin-Ciocalteu reagent, and then allowed the mixture to stand for 15 minutes. Next, we added 2mL of saturated sodium carbonate (2%) and let the mixture sit at room temperature for 30  minutes. To evaluate TPC, we used gallic acid as a standard and measured it with a spectrophotometer (Labo America, USA) at 760 nm. The total phenol content was ex-  pressed as mg of gallic acid per g of sample, using the linear regression equation ob-  tained from the calibration curve of standard gallic acid (y= 0.0002x – 0.0018)”.  

  2.5.2. Total flavonoid content (TFC) of Algae Extracts

No need to give details of already well-established methods. Relevant reference is enough. Delete the following paragraph, it is unnecessary

“First, we mixed 1 mL of the sample with 4 mL of water in a volumetric flask. Next, we added 0.75 mL of 5% sodium nitrite and 0.150 mL of 10% aluminum chloride to the mixture. After allowing it to sit for five minutes at room temperature, we added 0.5 mL of 1 M Sodium hydroxide. The TFC values were evaluated using a T80 UV/VIS spectrophotometer (PG Instrument Ltd., UK) at 510 nm. To express the results, we used the standard catechol calibration curve (y = 0.2462x + 0.2045) to determine the mg catechol equivalent per gram of the sample”.

Tables: Please add p-value in all tables indicating level of significance. Please remove the lines in the table. In the current form, table do not look attractive.

Author Response

Reviewer 1

The manuscript Foods-2547414 entitled " Utilization of Algae Extracts as Natural Antibacterial and Antioxidants for Controlling Foodborne Bacteria in meat products”

Overall, the paper is well written and very well explained.

I have few suggestions for improvement.

Why the authors did not study lipid oxidation by measuring TBARS values. It was very important to determine the oxidative stability of the meat products treated with algae extracts. Additionally, TVBN is also an important indicator in such studies missed by the authors. I will recommend to add the relevant data, if its available. It will improve quality.   

Response: Thank you very much for your accurate comment, unfortunately, right now we don’t have any relevant data related to TBARS and/or TVBN values, due to, our aim is mainly focuses on the cytotoxicity, antibacterial and antioxidant (via Diphe-nyl-1-picrylhydrazyl (DPPH) radical scavenging capacity) potentials of algae extracts, as well on the shelf life of the fortified meat. On the other hands, we will add these interesting parameters in the conclusion as future perspective in the upcoming studies.

Abstract: Please modify abstract according to the Journal’s guidelines. It should be in a single paragraph without sub headings. The main content of the abstract should include the briefly purpose of the research, the principal result and major conclusion. The abstract, in the present form, is not adequate. Please add specific results here not generalized. Additionally, (i) the authors must state the revised justification in the abstract to support the study; (ii) Conclusion in the abstract is too general, authors shall state the important finding in this study before going into the general application

Response: Thanks for your valuable comment, the abstract modified.

Introduction: The introduction section does not explain what was done on topic analyzed. Please include the relevant information about the algae extract antimicrobial effects specifically related to its use in meat products.

Response: Thanks for your important comment, the introduction improved.

Figure 1: Please improve the figure pixels and quality. Also remove background to improve their quality.

Response: Thanks for your important notice, we improved the resolution and the quality of figure 1 accordingly.

2.5.1. Total Phenolic content (TPC)

No need to give details of already well-established methods. Relevant reference is enough. Delete the following paragraph, it is unnecessary

“To begin, we added 0.1 mL of reconstituted extract to 0.1 mL of Folin-Ciocalteu reagent, and then allowed the mixture to stand for 15 minutes. Next, we added 2mL of saturated sodium carbonate (2%) and let the mixture sit at room temperature for 30  minutes. To evaluate TPC, we used gallic acid as a standard and measured it with a spectrophotometer (Labo America, USA) at 760 nm. The total phenol content was ex-  pressed as mg of gallic acid per g of sample, using the linear regression equation ob-  tained from the calibration curve of standard gallic acid (y= 0.0002x – 0.0018)”. 

Response: Thanks for your excellent notice, the paragraph removed.

  2.5.2. Total flavonoid content (TFC) of Algae Extracts

No need to give details of already well-established methods. Relevant reference is enough. Delete the following paragraph, it is unnecessary

“First, we mixed 1 mL of the sample with 4 mL of water in a volumetric flask. Next, we added 0.75 mL of 5% sodium nitrite and 0.150 mL of 10% aluminum chloride to the mixture. After allowing it to sit for five minutes at room temperature, we added 0.5 mL of 1 M Sodium hydroxide. The TFC values were evaluated using a T80 UV/VIS spectrophotometer (PG Instrument Ltd., UK) at 510 nm. To express the results, we used the standard catechol calibration curve (y = 0.2462x + 0.2045) to determine the mg catechol equivalent per gram of the sample”.

Response: Thanks, the paragraph deleted.

Tables: Please add p-value in all tables indicating level of significance. Please remove the lines in the table. In the current form, table do not look attractive.

Response: Thanks, we added P value in the tables.

Reviewer 2 Report

Dear authors,

I have carefully reviewed your article entitled "Utilization of Algae Extracts as Natural Antibacterial and Antioxidants for Controlling Foodborne Bacteria in Meat Products." While I find the topic interesting, I must express my opinion that the results are currently quite preliminary, and as presented, the article lacks scientific relevance.

Here are my comments, organized in the order of the article:

1.    HCV: The first time an acronym is used, it should be written in full.

2.    Abstract: The abstract does not follow the typical structure of a scientific text. I suggest referring to abstracts from other published articles in Foods to follow a more standard format.

3.    Supplementary Information (SI): All figures and tables that are already available in the main article should not be repeated in the SI. Generally, the SI is reserved for additional figures, tables, and relevant information beyond what is presented in the main article. In its current form, the SI does not add value.

4.    Table 2: It is unclear why Table 2 is repeated. Please clarify and indicate in the table header that the numbers correspond to the size of the inhibition zone (mm).

5.    Antioxidant activity: It is surprising that the antioxidant activity of the seaweed extracts studied is very insignificant. Can you provide any reasoning or comparative studies to support these findings? Additionally, be cautious about the use of different units in Table 2, where mg/mL and microgram/mL are used interchangeably.

6.    Table 7: The data in Table 7 are difficult to follow. Consider presenting these data in the SI and providing graphical representations in the main article for better clarity.

7.    Sensory analysis: The similarity of data between the control and different treatments in the sensory analysis raises questions about the experience and qualifications of the testers. Ensure that the testers are genuinely experienced and that their qualifications are well-documented.

8.   Some references related with the use of algae in food preservation is missing. check them!. And check the format of the references.

9.    Results and Discussion: The most critical improvement required from a scientific perspective is the lack of discussion in the "Results and Discussion" section. The section only presents the results, but no thorough discussion has been included, except in the case of antibacterial activity. Please provide a comprehensive discussion of the data, using the results of the study on phenolic compounds and flavonoids to support your discussion.

10.    Conclusions: In the conclusions, you mention "the possibility of using all three algae as food preservatives". However, as the study has only been completed with Padina pavonica, this statement is not accurate. Please correct this inconsistency. Furthermore, clarify the statement about the safety of the extracts for human consumption, considering the discrepancy with the statement made in line 360 of the manuscript where it is mentioned that the extracts are not entirely safe.

In summary, in addition to improving the presentation of figures and tables, it is crucial to rewrite the "Results and Discussion" section and the "Conclusions" section more comprehensively to ensure the article meets the requirement.

Sincerely,

Author Response

Reviewer 2

Dear authors,

I have carefully reviewed your article entitled "Utilization of Algae Extracts as Natural Antibacterial and Antioxidants for Controlling Foodborne Bacteria in Meat Products." While I find the topic interesting, I must express my opinion that the results are currently quite preliminary, and as presented, the article lacks scientific relevance.

Here are my comments, organized in the order of the article:

  1. HCV: The first time an acronym is used, it should be written in full.

Response: Thanks a lot for your careful reading of the manuscript, the whole word written as well.

  1. Abstract: The abstract does not follow the typical structure of a scientific text. I suggest referring to abstracts from other published articles in Foods to follow a more standard format.

Response: Thanks for your valuable comment, the abstract modified.

  1. Supplementary Information (SI): All figures and tables that are already available in the main article should not be repeated in the SI. Generally, the SI is reserved for additional figures, tables, and relevant information beyond what is presented in the main article. In its current form, the SI does not add value.

Response: Thanks for your important comment, the Supplementary Information removed.

  1. Table 2: It is unclear why Table 2 is repeated. Please clarify and indicate in the table header that the numbers correspond to the size of the inhibition zone (mm).

Response: Thanks for your accurate words, the heading modified, while Table 2 is for the 3 tested algae, while table 3 is only for the strongest algae which was Padina pavonica with MIC.

  1. Antioxidant activity: It is surprising that the antioxidant activity of the seaweed extracts studied is very insignificant. Can you provide any reasoning or comparative studies to support these findings? Additionally, be cautious about the use of different units in Table 2, where mg/mL and microgram/mL are used interchangeably.

Response: Thanks, no, we evaluated and discussed the antioxidant power in section 3.5. Antioxidant activity and DPPH radical Scavenging Capacity as well in Table 5.

  1. Table 7: The data in Table 7 are difficult to follow. Consider presenting these data in the SI and providing graphical representations in the main article for better clarity.

Response: Thanks a lot, the table modified.

  1. Sensory analysis: The similarity of data between the control and different treatments in the sensory analysis raises questions about the experience and qualifications of the testers. Ensure that the testers are genuinely experienced and that their qualifications are well-documented.

Response: Thank you very much, we add the ethical statement in the manuscript.

  1.  Some references related with the use of algae in food preservation is missing. check them!. And check the format of the references.

Response: Thank for your consideration, references modified as well.

  1. Results and Discussion: The most critical improvement required from a scientific perspective is the lack of discussion in the "Results and Discussion" section. The section only presents the results, but no thorough discussion has been included, except in the case of antibacterial activity. Please provide a comprehensive discussion of the data, using the results of the study on phenolic compounds and flavonoids to support your discussion.

Response: Thank very much, this section modified.

  1. Conclusions: In the conclusions, you mention "the possibility of using all three algae as food preservatives". However, as the study has only been completed with Padina pavonica, this statement is not accurate. Please correct this inconsistency. Furthermore, clarify the statement about the safety of the extracts for human consumption, considering the discrepancy with the statement made in line 360 of the manuscript where it is mentioned that the extracts are not entirely safe.

Response: Thank very much, the conclusion corrected and modified.

In summary, in addition to improving the presentation of figures and tables, it is crucial to rewrite the "Results and Discussion" section and the "Conclusions" section more comprehensively to ensure the article meets the requirement.

Sincerely,

Response: Thank a lot, the sections corrected and modified.

Reviewer 3 Report

It is an interesting study on alternative sources of antioxidants and antimicrobials.

However, it has some observations.

1.- Please improve the tables, follow the authors' guide, and look disproportionate to the text.

2.- The use of units of measurement such as minutes and hours, please check in the author's guide if the complete word is used or if it is used (h, min, s, etc.).mg/mL is placed next to the quantities, please correct this and so on in several quantities.3.-Check the scientific names and correct those that are not spelled correctly.

4.-It is important to discuss the relationship between the content of phenolic compounds, including flavonoids in this point 3.7. Shelf life of meat fortified with Padina pavonica extract, why does it inhibit? Because they are more effective against some strains and not others. The discussion should be deeper. le it seems to be understood that it is the antioxidant and antimicrobial activity of the extracts in the hamburger, in the results it does not comment on the antioxidant activity. Was it evaluated? If not, a title change should be considered.

5.- It is important that you use a similarity program of your choice.

5.Check the spelling, and grammar of your entire manuscript, especially the lack of spacing between words or numbers of words.

Author Response

Reviewer 3

It is an interesting study on alternative sources of antioxidants and antimicrobials.

However, it has some observations.

1.- Please improve the tables, follow the authors' guide, and look disproportionate to the text.

Response: Thanks a lot for your careful reading of the manuscript, tables modified.

2.- The use of units of measurement such as minutes and hours, please check in the author's guide if the complete word is used or if it is used (h, min, s, etc.).mg/mL is placed next to the quantities, please correct this and so on in several quantities.

Response: Thank you very much, observed.

3.-Check the scientific names and correct those that are not spelled correctly.

Response: Thanks for your excellent note, corrected.

4.-It is important to discuss the relationship between the content of phenolic compounds, including flavonoids in this point 3.7. Shelf life of meat fortified with Padina pavonica extract, why does it inhibit? Because they are more effective against some strains and not others. The discussion should be deeper. le it seems to be understood that it is the antioxidant and antimicrobial activity of the extracts in the hamburger, in the results it does not comment on the antioxidant activity. Was it evaluated? If not, a title change should be considered.

Response: Many thanks for your valuable notice, modified the discussion, and the antioxidant activity Thanks, no, we evaluated and discussed the antioxidant power in section 3.5. Antioxidant activity and DPPH radical Scavenging Capacity as well in Table 5.

5.- It is important that you use a similarity program of your choice.

Response: Thanks for your excellent note, similarity program used and the manuscript plagiarism modified.

Comments on the Quality of English Language

  1. Check the spelling, and grammar of your entire manuscript, especially the lack of spacing between words or numbers of words.

Response: Thanks for your kind comment, corrected.

Figure 1: Please improve the figure pixels and quality. Also remove background to improve their quality.

Response: Thanks for your important notice, we improved the resolution and the quality of figure 1 accordingly.

Tables: Please add p-value in all tables indicating level of significance. Please remove the lines in the table. In the current form, table do not look attractive.

Response: Thanks, we added P value in the tables.

1.- Please improve the tables, follow the authors' guide, and look disproportionate to the text.

Response: Thanks for your accurate comment, we improved the quality of the tables accordingly.

In the results it does not comment on the antioxidant activity. Was it evaluated? If not, a title change should be considered.

Response: Thanks, no, we evaluated and discussed the antioxidant power in section 3.5. Antioxidant activity and DPPH radical Scavenging Capacity as well in Table 5.

Round 2

Reviewer 1 Report

I am satisfied with authors reply

Author Response

Dear Reviewer 1, thanks for your kind efforts to fulfill this manuscript

Regards

Reviewer 2 Report

Dear authors,

Firstly, it has been challenging to carry out the review of this new version of the article due to the attachment of a tracked changes PDF file. It would have been better to use a Word file or to have included a second PDF file with the new version.

Apart from this, upon re-reading, it is evident that the authors have not conducted a rigorous revision, neither in the new presentation of the article nor in their response to the reviewers. For instance, regarding the limited antioxidant activity of the extracts, I suggested to provide a rationale or conduct a comparative study to support the obtained data. However, the only response provided was, "Thanks, no, we evaluated and discussed the antioxidant power in section 3.5. Antioxidant activity and DPPH radical Scavenging Capacity as well in Table 5." Clearly, there has been no discussion of the results, only a presentation of data.

In a broader sense, I also suggested that the "Results and Discussion" section only contained results and I suggested to enlarge this section with a real dicussion. Unfortunatelly, It was also neglected to do. Upon reading the new version, it is noticeable that there are hardly any differences between the deleted paragraphs and the newly written ones. Similarly, in the references, the DOI has been included, but most do not adhere to the journal's formatting.

Finally, in the conclusions, the initial paragraph is maintained and only a sentence where two acronyms "TBARS and TVBN" are introduced for the first time, which are out of context and without explanation.

Therefore, with these corrections in mind, my decision once again is "Reconsider after major revision."

Several typos are spreaded along the article. A careful english revision is needed.

Author Response

Dear authors,

Firstly, it has been challenging to carry out the review of this new version of the article due to the attachment of a tracked changes PDF file. It would have been better to use a Word file or to have included a second PDF file with the new version.

Response: Thank you very much for your kind efforts, sorry for inconvenience, this time I uploaded a revised version without track changes and the file with red color font of the corrected and added words.

Apart from this, upon re-reading, it is evident that the authors have not conducted a rigorous revision, neither in the new presentation of the article nor in their response to the reviewers. For instance, regarding the limited antioxidant activity of the extracts, I suggested to provide a rationale or conduct a comparative study to support the obtained data. However, the only response provided was, "Thanks, no, we evaluated and discussed the antioxidant power in section 3.5. Antioxidant activity and DPPH radical Scavenging Capacity as well in Table 5." Clearly, there has been no discussion of the results, only a presentation of data.

Response: Thank you very much for your accurate comments, corrected with red color font.

In a broader sense, I also suggested that the "Results and Discussion" section only contained results and I suggested to enlarge this section with a real dicussion. Unfortunatelly, It was also neglected to do. Upon reading the new version, it is noticeable that there are hardly any differences between the deleted paragraphs and the newly written ones. Similarly, in the references, the DOI has been included, but most do not adhere to the journal's formatting.

Response: Thank you very much, the discussion extended and the references corrected according to the journal guidelines, and more 42 relevant and recent until 2023 more references added plus the original 32 references so now the whole references are 74 ones.

Finally, in the conclusions, the initial paragraph is maintained and only a sentence where two acronyms "TBARS and TVBN" are introduced for the first time, which are out of context and without explanation.

Response: Thanks a lot, removed and edited.

Therefore, with these corrections in mind, my decision once again is "Reconsider after major revision."

Comments on the Quality of English Language

Several typos are spreaded along the article. A careful english revision is needed.

Response: Thank you very much, all done.

Reviewer 3 Report

Although the authors replied satisfactorily to what was observed, I do not understand why the document was uploaded to the platform with change control. Was it a mistake? Or an oversight? Please check this detail.

Author Response

Dear Reviewer 3

Greetings

Thanks for your great efforts, and sorry for inconvenience, this time I will upload a version of the manuscript without track changes and the correction and addition are with the red color

Regards